# A systemic approach to estimate and validate RP-HPLC assay method for remdesivir and favipiravir in capsule dosage form

**Nasr Ullah Khan**, **Muhammad Iqbal**, **Barkat Ali Khan***

Gomal Center of Pharmaceutical Sciences, Faculty of Pharmacy, Gomal University, Dera Ismail Khan, Khyber Pakhtunkhwa, Pakistan

* barkat.khan@gu.edu.pk

## Abstract

The implementation of quality by design offer quality and safety product to patients, efficient processes for continuous improvements to manufacturers, negligible amount of batch failures and robust product quality attributes. This study was designed to establish a simple, specific, precise, and accurate reverse phase-high performance liquid chromatographic (RP-HPLC) method for the quantitative determination of remdesivir and favipiravir in Capsule dosage forms. The RP-HPLC method was performed on a Kromasil 100A C18 column (250 mm x 4.6 mm, 5 µm particle size) with a mobile phase containing (50 mL of acetonitrile, 350 mL methanol, 100ml of water and 0.5 mL of Phosphoric acid). The flow rate was 1.0 mL/min. The ultraviolet (UV) detection wavelength was 300 nm, and the column temperature was set at ambient. Linearity and range stock solutions were prepared as 50% to 150%. The calibration curves showed a good linear response ranged from 0.02 to 0.06 mg/ml (r=1.0000) for favipiravir and 0.022 to 0.066 mg/ml (r=1.0000) for remdesivir, and the average recoveries were 99.9% for favipiravir and 99.8% for remdesivir in assay test. The retention time of favipiravir was RT 11.5 minutes and remdesivir 20.95 minutes, tailing factor was not more than 1.5 and resolution of more than 2.0 respectively. The extended run time was supported by high concentration of favipiravir and uneven peak behavior of remdesivir in single run, which disturb resolution, retention times, injection repeatability. Reducing runtime on HPLC will lead to coelution of peaks, poor resolution, and loss of sensitivity during degradation profiling. This issue may be overcome with future UHPLC technique. The method was validated in accordance to International Council for Harmonization of Technical Requirements of Pharmaceuticals for Human Use (ICH) guidelines. The limit of detection (LOD) for Favipiravir was 0.104mg/ml and Remdesivir was 0.052mg/ml and limit of quantification (LOQ) for Favipiravir was 0.316mg/ml and Remdesivir was 0.158mg/ml respectively. In stress conditions, this product undergoes degradation and was considered sensitive to Acid, Alkali, Oxidation, reduction, hydrolysis, high temperature and humidity. The assay methods was simple, rapid, sensitive, repeatable, eco-friendly, stability indicating and can be used cost-effectively for the testing of these two drug substances in capsule dosage form. Due to the emergent need of

**Data availability statement:** All relevant data are within the manuscript and its Supporting Information files.

**Funding:** The author(s) received no specific funding for this work.

**Competing interests:** The authors have no competing interest.

antivirals for flu management, this coloaded combination was considered as dire need of society. Also, we reported that the product must be stored under ambient temperature, low humidity, and protected from light exposure.

## Introduction

Test method development is the determination of a set of experimental method to check/assess analytical procedure in chemical compounds. These analytical parameters are used to identify, segregate, quantify, and to analyze the chemical constituents in drug products suitable for commercial scale up manufacturing [1,2]. The main focus of test method development is to deliver a perfect background for achievement of consistent results without any interruption or delays during commercial scale up batches. If conditions are altered in any case, these slight variations are covered in test method validated parameter to reproduced and robust the analytical method from any interfering method conditions. Test method development is helpful to calculate and measure the critical method parameters and to eliminate their impact on product quality, quantity and precision. Test methods used for any purpose must be reliable, accurate, precise, robust, stable, configurable for any GLP or GMP premises and must be validated according to ICH Q2(R1). Test method validation ensure us to authenticate the analytical test method for a variety of multiple concentrations so that any alteration in product formulation or change in concentration of any ingredient do not involve supplementary validation. Once the parameters of the method have been established, controlled, qualified and validated, the influence of these results on out of specification for a particular case or process suitability needs to be standardized or qualified for established parameter or process [3].

A validated UHPLC method coupled with mass spectrophotometer for remdesivir and favipiravir was available with PMID: 37228397, but this method has limitations of LCMS technology not available frequently in pharmaceutical laboratories. Validation parameters discussed have the limitations of broader limits for selectivity ± 20%, linearity correlation 0.98, Accuracy ± 20%, carryover ± 20%. A combination of acetonitrile and formic acid used for mobile phase with internal standard for stabilization and single peak elution was shown in chromatogram instead of both peak elution in one single run [4].

A recent article has been published with DOI # 10.1002/cem.3548, the mobile phase selection was acetonitrile, buffer pH 2.9 and triethylamine. In this combination triethylamine creates significant selectivity changes over time, resolution issue, stability issues, abnormal peaks, column damage and cannot be used in commercial batches analysis. This method has demonstrated recoveries of 99–106% while our method has recoveries of 98–100%, precision and robustness with RSD not more than 1.4% while in our validated method the RSD is not more than 0.5% for all these parameters. One another limitation is the use of Remdesivir IV form which has short half life, while we use Remdesivir oral nucleoside (GS-621763) [5].

Currently, for capsule dosage form, this oral combination was found to be most effective on the basis of all pharmacopeial parameters studied. The RP-HPLC method C18 column with a mobile phase was used for assay and dissolution test for this combination which is entirely different from single drug substances studied on UV or HPLC technologies. Herein, we aimed at establishing a reliable reverse phase-high performance liquid chromatographic (RP-HPLC) method for the quantitative determination of Favipiravir and Remdesivir in Capsule dosage forms, following ICH guidelines [6].

Method Operable Design Region (MODR) in Quality by Design (QbD) is a key concept for development of pharmaceuticals. It ensures consistent product quality and define the range of critical process parameters like; solvent selection ratio, reaction temperature and

time, pH control, catalyst and mixing speed. Critical material attributes like peak purity, particle size distribution, polymorphic form and optimization for scale up includes; continuous batch monitoring, recoveries and environmental impact, robustness effect. In this study, model validation follows a systematic approach under Quality by Design (QbD) principles. Data collection was analytically verified through ICH guidelines, internal validation was assessed through correlation coefficient and method scaleup was measured through robustness parameters [7].

The components to be tested in liquid chromatography to be eluted and analyzed are carried in a small volume, i.e., microliters, with the flow of mobile phase which saturate the column during analysis. The components which are under test, elutes through the column at different flow rates, this interaction with a stationary phase is due to specific adsorbent characteristics of embedded silica and carbon chain. The flow and pace of each eluting component in mixture is mainly dependent on its chemical nature and characteristic of the stationary phase or column and also on the solvent mix composition of the mobile phase. The particular time by which a definite constituent elutes is called its retention time. There are many types of columns which are packed with adsorbents materials of varying size, porosity and in the chemical nature of their surface chemistry. The particle size plays a pivotal role as smaller the particle size of stationary phase materials, greater operational pressure ("backpressure") is required to achieve better chromatographic resolution. Adsorbent materials may be hydrophobic or hydrophilic in nature [8]. The chemical mixture also called as composition of the mobile phase may be either constant "isocratic mode" or changed "gradient mode" as per requirement of eluting substance during chromatographic analysis. Isocratic mobile phase is mostly used for the separation of mixtures of components that are nature wise differ in their adsorbent characteristic for the stationary phase. While in gradient elution the constituents are separated on the basis of low to high affinity for the mobile phase solubility characteristic and then adsorb at different time interval on stationary phase. Thus, the strength of the mobile phase mixture depends upon the constituent mixture retention times, as the high eluting molecules will be carried faster through the stationary phase and have shortest possible retention times. In most of the cases for gradient profile in reversed phase chromatography, the organic solvent composition starts from 5% in water or aqueous buffer and on linear increment it reaches up to 95% acetonitrile over 5–55 minutes depending upon total run time. Several starter runs are analyzed before sequence initiation during compound mixture elution in order to proceed with the best HPLC method which gives optimum separation and resolution [9].

Literature survey reveals that only few analytical methods either available as separate drug quantification and no combined HPLC method has yet being studied, so this will be novel method to elute both drug substances separately. The entire HPLC method is used to separate, identify, and quantify compounds within a mixture [10].

Favipiravir in its phosphoribosylated form (Favipiravir-RTP) is an RNA-dependent RNA polymerase inhibitor, which suppresses the RNA polymerase activity. It was approved in Japan as tablet in 2014 for the treatment of influenza infections, and since then, it has been authorized in several other countries for use against various viral diseases, including COVID-19 [11].

Remdesivir was the first anti-viral prodrug to be used for these kinds of viral infections, ongoing studies continue to assess its broader applications and effectiveness against different viral pathogens. In 2019, during COVID-19 in the United States, FDA approved remdesivir as emergency treatment for used in this manner, Intravenous Remdesivir can rapidly achieve therapeutic concentrations in the bloodstream within 24 hours, ensuring a more immediate and potent antiviral effect [12].

## Materials and methods

### Chemicals

Standard Favipiravir (Shaanxi Haibo Biotechnology Co. Ltd. China), Standard Remdesivir (Anhui Haikang Pharmaceuticals Co. Ltd. China), Distilled Water (Novamed Pharmaceuticals Lahore), Methanol HPLC grade (Merck, Germany), Acetonitrile HPLC grade (Merck, Germany), Phosphoric Acid (Sigma Aldrich, Germany).

### Instruments

Analytical weighing balance (Sartorius, Japan), Pipette (Pyrex IWAKI, Indonesia), Ultra Sonicator (Elma, Germany), Filtration assembly (Sartorius, Germany), High Performance Liquid Chromatography with DAD Detector (Shimadzu Corporation, Japan), Filter cellulose nitrate (Sartorius pore size 0.45um), Whatman Filters (GE Healthcare Life Sciences, USA), Beakers (Pyrex, IWAKI, Indonesia). Volumetric flasks (Pyrex IWAKI, Indonesia).

### Chromatographic conditions

- Column: 250 mm × 4.6 mm, 5 μm, C18

- λ max: 300 nm

- Temperature: Ambient

- Flow Rate: 1 ml/ minute

- Inj. Volume: 10 μL

- Mobile phase consists of a mixture of acetonitrile (50 ml): methanol (350 ml): water (100 ml) and 0.5 ml phosphoric acid.

### Procedure

Took twenty capsules and weigh, then collect the shell content carefully and pour into pestle and mortar. Took crushed powder equivalent to 1 capsule weight, i.e., 550 mg and dissolve it in 40 ml of mobile phase in 100 ml flask, dilute it up to the mark with the same diluent. Sonicate the solution for approximately 10 minutes. Filter the solution through membrane filter millipore 0.45 microns, then pipette 1 ml of the filtrate in 25 ml volumetric flask, dilute with diluent to volume. The final concentration was 0.04 mg/ml of remdesivir & 0.08 mg/ml of favipiravir respectively. The average amount of drug content per capsule that was present in each batch was calculated with the help of suitable calibration curve obtained from the area of reference standard solution [13].

**Preparation of the standard solution.** Weigh accurately 25 mg of remdesivir & 50 mg of favipiravir into 25 ml volumetric flasks. Add mobile phase to dissolve and sonicate for 5 minutes and make up the final volume to 25 ml. Pipette 1ml from standard stock solution in 25 ml volumetric flask, dilute with mobile phase to volume. The solution contains 0.04 mg/ml of remdesivir & 0.08 mg/ml of favipiravir.

**Preparation of the test sample.** Weigh powder equivalent to weight of one capsule containing 100 mg of remdesivir and 200 mg of favipiravir in 100 ml volumetric flask, dissolve in mobile phase. Sonicate the solution for 10 minutes. Filter the solution through membrane filter, millipore, 0.45 microns, pipette 1 ml of the filtrate in 25 ml volumetric flask, dilute with mobile phase to volume. Final concentration was 0.04 mg/ml of remdesivir and 0.08 mg/ml of favipiravir. Inject separately 10 μl of standard and sample solutions and record peak

responses. Calculate the content of remdesivir and favipiravir in the capsules by the following formula:

Percent of remdesivir and favipiravir release =Au/As× Rs/Ru × Pot. of std [14]

Where, Au and As were peak areas of sample and standard.

Rs and Ru were dilution factors of standard and sample respectively.

**RP-HPLC method development.** To optimization process for the HPLC method development for favipiravir and remdesivir initially attempts were made by using various solvent systems and column configurations (C8 and C18) were insufficient to achieve chromatographically sound separation and retention for both drug substances. The best retention and separation profile were achieved with the mobile phase consisting of Acetonitrile: Methanol: Water in a ratio of 50:350:100, along with 0.5 mL phosphoric acid. This mobile phase effectively facilitated the elution of both analytes on stainless steel kromasil 100A column 4.6×250mm, 5μ- C18, merck at wavelength of 300 nm with a flow rate of 1 ml/ minute. The principle of *"like dissolves like"* was employed to ensure analyte stability in the selected solvent. Among potential options, the mobile phase was selected as the diluent for consistency and compatibility across the test method validation. The shortlisted mobile phase composition and column combination provided the robustness required for validation of test parameters such as linearity, precision, accuracy, and specificity [15]. Test method HPLC parameters include; Specificity, Solution Stability, Linearity, Range, Precision, Accuracy, Robustness, Limit of Detection, Limit of Quantification, Stress Degradation and System Suitability. Each parameter should be meticulously documented, including raw data, calculations, and results for internal validation protocols [16].

## Result and discussion

### Specificity

Specificity shows that the procedure is unaffected by the presence of impurities or excipients. Specificity was performed by running a standard solution (as identification test), sample solution, blank, mobile phase comparing with a placebo run and run time should be at least in triplicate of principle peak [17]. The data is depicted in Table 1.

No peak detected for mobile phase, blank and placebo at RT 11.5 and 20.95 minutes respectively (as shown in Fig 1a and 1b) means no interference observed at specified retention time.

### Solution stability

The test solution was prepared and a portion of solution was stored at room temperature (25°C) and one sample at 2–8°C. Test solution was analyzed after 24 hours, 48 hours and 72 hours and same for refrigerated sample was analyzed as per procedure given under experimental condition and compared with freshly prepared reference standard [18]. The % difference in the area of test solution at each interval was calculated and recorded. The data has been shown in Table 2 while the raw data is presented in S1 and S2 Tables.

Table 1. Specificity test results.

| Sr. No. | Name of Solution | Retention Time (min) |
|---|---|---|
| 1 | Blank | No peak detected |
| 2 | Placebo | No peak detected |
| 3 | Favipiravir drug substance | 11.49 minutes |
| 4 | Remdesivir drug substance | 20.95 minutes |
| 5 | Run Time | 32 minutes |

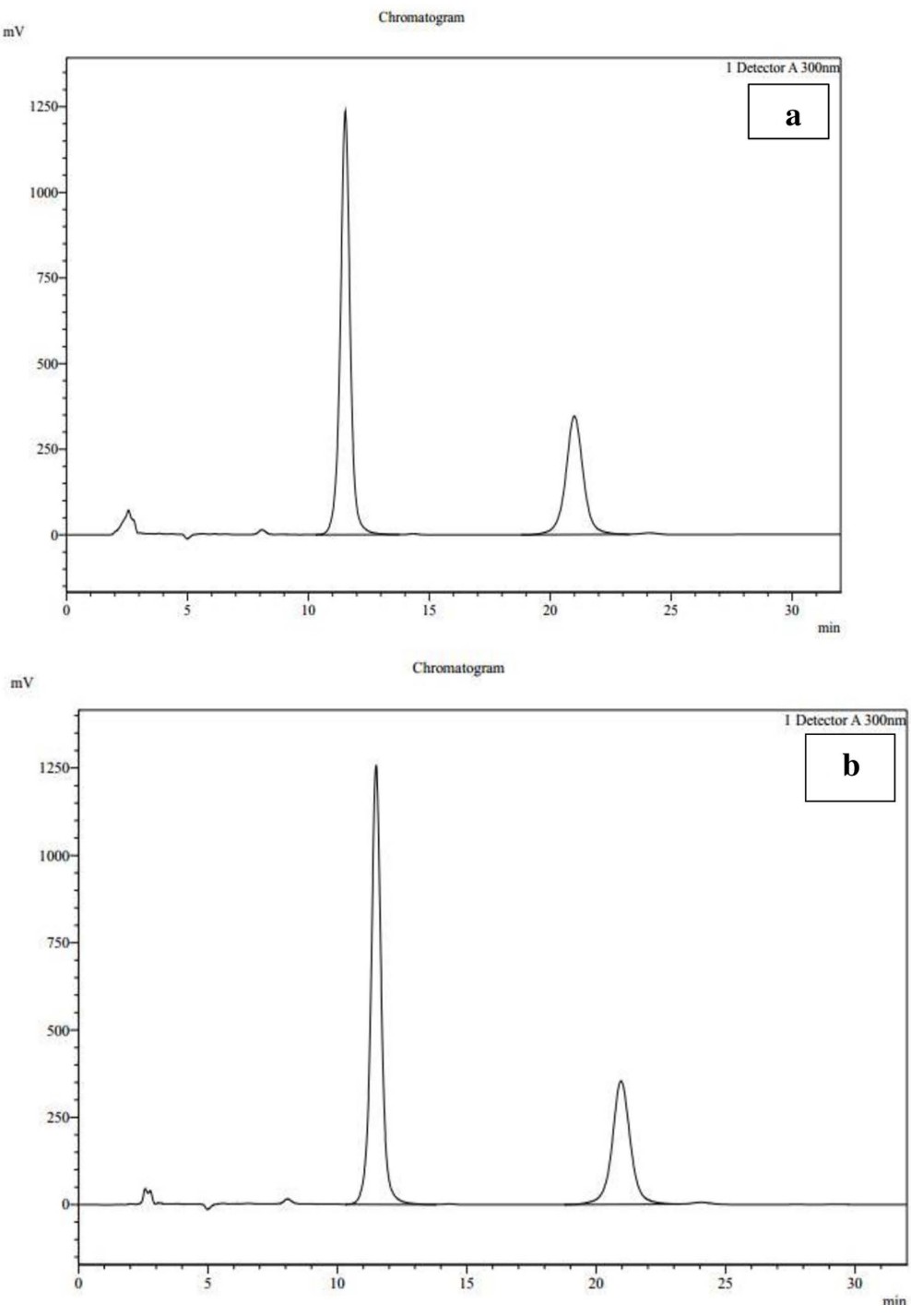

**Fig 1. a: Chromatogram of standard, b: Chromatogram of sample.**

## Linearity and range

The concentrations of the standard solution are varied across specific percentages of the target concentration, which are 50%, 75%, 100%, 125%, and 150% of the target value. This helps in confirming that the HPLC method is capable of providing accurate responses across a range of concentrations. For each concentration, a solution is injected into the HPLC system, and the response (peak area) at 300 nm is recorded. The peak area is directly related to the amount of the compound present in the sample. A graph is created with concentration on the X-axis and peak area on the Y-axis. A straight line indicates a consistent, predictable response from the instrument, which is crucial for accurate quantification in HPLC analysis. The correlation coefficient (Not less than 0.999), y-intercept, slope of the regression line and residual sum of squares was calculated and reported [19]. The correlation coefficient was 0.9999 which was well within acceptance limit of Not Less Than (NLT) 0.999, y-intercept, slope of the regression line and residual sum of squares was calculated. % RSD of triplicate assay at each level was less than 2.0%, all value (as shown in Figs 2,3) complies with acceptance limits and a linear response was observed at range of 50–150% [20]. Raw data is presented in S3 Table.

## Precision

Precision of an analytical procedure refers to the consistency and repeatability of the results when the same sample is analyzed multiple times under the same conditions. It is often evaluated by comparing the results of multiple measurements of the same sample to determine how closely the individual results agree with each other. A different analyst carried out the analysis on different day, using different HPLC system. The % of relative standard deviation, confidence interval at 90% and the absolute difference between the mean results obtained from the

**Table 2. Solution stability with their absolute difference at various time intervals.**

| Parameter | % Recovery of Favipiravir | % Deviation Favipiravir | % Recovery of Remdesivir | % Deviation Remdesivir |
|---|---|---|---|---|
| Initial | 99.88% | 0.11% | 99.75% | 0.24% |
| 25C-24H | 97.63% | 2.31% | 96.97% | 3.04% |
| 4C-24H | 99.18% | 0.70% | 98.93% | 1.06% |
| 25C-48H | 94.35% | 5.86% | 93.49% | 6.52% |
| 4C-48H | 96.75% | 3.23% | 95.95% | 4.05% |
| 25C-72H | 86.77% | 15.11% | 83.71% | 16.33% |
| 4C-72H | 93.76% | 6.53% | 92.57% | 7.45% |

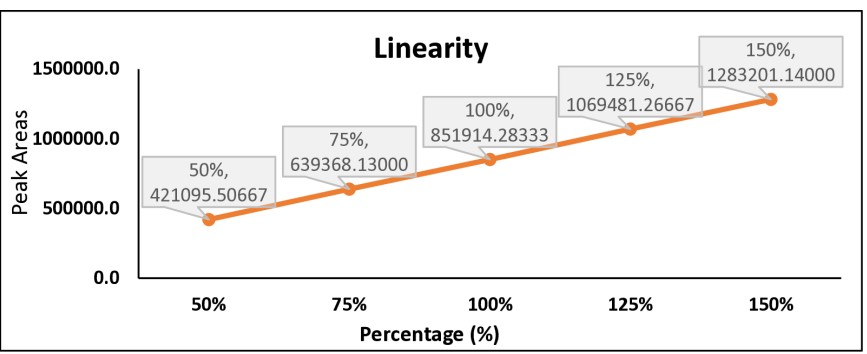

**Fig 2. Favipiravir linearity graph.**

repeatability analysis and the result of intermediate precision was calculated. The % RSD for six replicates should not be more than 2.0% [21]. The data is depicted in Table 3 while the raw data has been presented in S4 and S5 Tables.

## Accuracy

A measure of exactness of the analytical method. The method is said to be accurate if the method provides the true answer. For this purpose, three replicates of standard solution of 100% (favipiravir 0.08 mg/ml and remdesivir 0.04 mg/ml respectively) was injected and test solution, i.e., 80%, 100% and 120% was prepared as spiked assay in triplicate runs and tested results were to be tabulated as mean recovery and shown in Table 5. The raw data can be traced in S8 and S9 Tables.

All the individual recoveries for favipiravir and remdesivir were found well within 97.0% and 103.0%. Mean recovery was 99.76% for favipiravir and 99.92% for remdesivir, which was well within specified limits 98.0% to 102.0%. 90% confidence interval was 0.0034 for favipiravir and 0.0028 for remdesivir, all parameter complies well within specified limits [22].

## Limit of detection (LOD) and limit of quantitation (LOQ)

The obtained LOD results from linearity curve for favipiravir was 0.104 mg/ml and 0.052 mg/ml respectively, The obtained LOQ results from linearity curve for favipiravir was 0.316 mg/ml

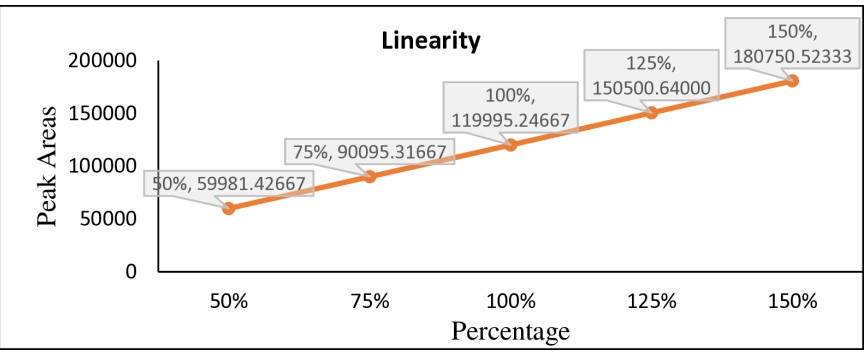

**Fig 3. Remdesivir linearity graph.**

Table 3. Result table for precision favipiravir.

| Description | Results % Analyst 1 | STDEV | Results % Analyst 2 | STDEV |
|---|---|---|---|---|
| Sample No. 1 | 99.47 | 0.432% | 99.99 | 0.278% |
| Sample No. 2 | 100.36 | | 100.15 | |
| Sample No. 3 | 99.84 | | 99.66 | |
| Sample No. 4 | 99.45 | | 99.66 | |
| Sample No. 5 | 100.21 | | 100.31 | |
| Sample No. 6 | 100.40 | | 100.19 | |
| **Confidence Interval @ 90%** | 0.2239 | – | 0.1442 | – |

For Analyst 1 Favipiravir: The mean was observed as 99.95%, relative standard deviation was 0.432% and confidence interval at 90% was 0.2239.

For Analyst 2 Favipiravir: The mean observed was 99.99%, relative standard deviation was 0.278% and confidence interval at 90% was 0.1442. This shows the repeatability results are well within limits for both analysts.

and 0.158 mg/ml respectively, so beyond these limits the concentrations will not be quantified easily for both drug substances [23]. A non-linear plot was observed due to higher concentration, which leads to detector saturation, altering response factor and non-homogenous sample. However, the method was precise, accurate, linear and robust for the parameters studied. These LOD and LOQ calculations were based on Standard Deviation of the response and slope of the calibration curve from linearity and precision test.

## Robustness

The robustness of an analytical procedure is a measure of its capacity to remain unaffected by typical variations that can be deliberately introduced to assess the robustness of an analytical procedure. In the case of liquid chromatography, examples of typical variations are:

(i) Mobile phase pH (ii) Lambda Max (λ-max) (iii) Column Variability (iv) Flow Rate

**Table 4. Result table for precision remdesivir.**

| Description | Results % Analyst 1 | STDEV | Results % Analyst 2 | STDEV |
|---|---|---|---|---|
| Sample No. 1 | 99.83 | 0.418% | 100.73 | 0.275% |
| Sample No. 2 | 100.58 | | 100.42 | |
| Sample No. 3 | 100.33 | | 100.00 | |
| Sample No. 4 | 99.63 | | 100.49 | |
| Sample No. 5 | 99.74 | | 100.74 | |
| Sample No. 6 | 100.52 | | 100.34 | |
| **Confidence Interval @ 90%** | 0.2181 | – | 0.143 | – |

For Analyst 1 Remdesivir: The mean observed was 100.10%, relative standard deviation was 0.418% and confidence interval at 90% was 0.2181.

For Analyst 2 Remdesivir: The mean observed was 100.45%, relative standard deviation was 0.275% and confidence interval at 90% was 0.1430. This shows the repeatability results are well within limits for both analysts. Data for precision remdesivir is presented in Table 4 while the raw data has been presented in S6 and S7 Tables.

**Table 5. Tabulated results for accuracy.**

| Sr. # | Name of Solution | % Assay Favipiravir | % Recovered Favipiravir | % Assay Remdesivir | % Recovered Remdesivir |
|---|---|---|---|---|---|
| 1 | **60% Concentration** | 59.41% | 99.02% | 59.60% | 99.34% |
| | | 59.35% | 98.92% | 59.53% | 99.22% |
| | | 59.38% | 98.96% | 59.61% | 99.32% |
| 2 | **80% Concentration** | 79.32% | 99.15% | 80.08% | 100.11% |
| | | 79.86% | 99.82% | 79.84% | 99.80% |
| | | 79.71% | 99.64% | 80.41% | 100.52% |
| 3 | **100% concentration** | 99.96% | 99.96% | 99.94% | 99.94% |
| | | 100.31% | 100.31% | 99.88% | 99.88% |
| | | 99.74% | 99.74% | 100.37% | 100.37% |
| 4 | **120% Concentration** | 120.47% | 100.39% | 120.01% | 100.01% |
| | | 120.28% | 100.24% | 119.97% | 99.98% |
| | | 120.13% | 100.11% | 120.54% | 100.45% |
| 5 | **140% Concentration** | 140.51% | 100.37% | 140.03% | 100.02% |
| | | 139.63% | 99.74% | 140.08% | 100.05% |
| | | 140.08% | 100.06% | 139.82% | 99.87% |
| **Confidence Interval @ 90%** | | 0.0034 | | 0.0028 | |

The absolute difference or (assay results obtained from normal and the altered method should not be more than 2.0%). If a failure is observed, that parameter should be highlighted as a critical parameter. The obtained data has been shown in Table 6. The Raw data is presented in S10 and S11 Tables.

Influence of variations in mobile phase with respect to solvent ratios ±5% (Results shows significant variations, hence variations in mobile phase change was considered as critical parameter means significant change). Influence of variations in lambda max, results has shown variation with ± 3 nm, hence wavelength change was also considered as critical parameter. Variation in columns with different configuration has no impact on results, i.e., A stainless-steel column 4µm C18, 150 mm in length instead of stainless-steel column 5µm C18, 250 mm in length, hence change of column size and particle size has no impact on results. Flow rate results shows variation within ± 10%, Hence change in flow rate will alter retention time and was considered as critical parameter [24].

## Stress degradation

A stability-indicating method is a critical analytical tool used during the drug development phase to ensure the long-term stability and quality of a pharmaceutical product. Prepare in triplicate samples solutions and standard solution according to the analytical method. Analyzed all samples according to procedure on the either the same day after treatment with different stress conditions mentioned below in Table 7 [25]. Raw data can be traced in S12 and S13 Tables.

Peak Purity: Peak purity calculated was 99.71%, which was well within limit of 0.99. Temperature effect results as at 70 °C exposure of samples, the results were declined 23.63%, hence considered as critical parameter. Temperature and humidity also effect results, a decline of 11.77% was observed, hence considered as critical parameter. Light has affected results which are approximately 4.0%, hence considered as crucial parameter. Acid treatment has affected results up-to 17.90%, hence considered as critical parameter. Base treatment has also affected

**Table 6. Result data for robustness.**

| Parameter | % Recovery Favipiravir | % Deviation | % Recovery Remdesivir | % Deviation |
|---|---|---|---|---|
| +0.2 mL | 110.16% | −9.221% | 110.36% | −9.385% |
| −0.2 mL | 150.18% | −33.413% | 105.32% | −5.055% |
| + 3 nm | 85.22% | 17.342% | 85.40% | 17.093% |
| −3 nm | 80.10% | 24.847% | 80.13% | 24.805% |
| 5% MP (+) | 88.57% | 12.904% | 88.36% | 13.167% |
| 5% MP (−) | 92.28% | 8.367% | 92.19% | 8.474% |
| Column 4µm C18, 150mm change | 99.52% | 0.48% | 99.41% | 0.51% |

**Table 7. Stress degradation for multiple parameters.**

| Parameter | % Recovery Favipiravir | % Deviation | % Recovery Remdesivir | % Deviation |
|---|---|---|---|---|
| 70 C | 80.89% | 23.625% | 80.88% | 23.63% |
| 40 C & RH 75% | 90.07% | 11.020% | 89.46% | 11.77% |
| Acid Treatment | 85.06% | 17.564% | 84.81% | 17.90% |
| Base Treatment | 88.32% | 13.220% | 88.36% | 13.17% |
| Peroxide Treatment | 93.97% | 6.824% | 93.31% | 7.57% |
| 1.2 M Lux | 96.42% | 4.049% | 96.47% | 4.00% |

results and a decline of 13.17% observed, hence considered as critical parameter. Oxidation/ reduction treatment has affected results and a decline of 7.57% observed, hence considered as critical parameter [26].

## System suitability

This part of the test evaluates the performance of the system under the specific conditions, confirming the reliability of the analytical method. After six replicate injections at final concentration of 0.04 mg/ml of remdesivir & 0.08 mg/ml of favipiravir in both standard and sample respectively. Inject separately 10 µl of Standard and Sample solutions and record peak responses. Calculate the content of remdesivir and favipiravir in the capsules by the following formula:

Percent of remdesivir and favipiravir release = Au/As× Rs/Ru × Pot. Of Std [27]

For favipiravir: The number of theoretical plates and the symmetry factor was greater than 4000 and not more than 2.0 respectively. % RSD for standard was 0.38% which was well within limit of NMT 2.0%.

For Remdesivir: The number of theoretical plates and the symmetry factor was greater than 4000 and not more than 2.0 respectively. % RSD for standard was 0.33% which was well within limit of NMT 2.0%. Hence the system suitability test results for 6 replicates sample injections complies with specifications [28]. Raw data regarding system suitability is depicted in S14 and S15 Tables.

## Conclusion

The reported RP-HPLC method is a significant advancement due to its simplicity, rapidity, and reproducibility. The validation results show that it provides good precision, accuracy, and reliability, which are essential qualities for a robust analytical method. One of the key benefits of the method is the use of a straightforward mobile phase and an isocratic elution mode, which simplifies the overall process. This makes the method well-suited for routine use in quality control laboratories, such as immediate release capsules. Furthermore, this method novelty lies in the absence of previously reported stability-indicating methods for such dosage forms, making this approach unique and valuable in the context of stability testing and quality control.

## Method limitations

The extended run time was supported by high concentration of favipiravir and uneven peak behavior of remdesivir in single run, which disturb resolution, retention times, injection repeatability. Reducing runtime on HPLC will lead to coelution of peaks, poor resolution, and loss of sensitivity during degradation profiling. This issue may be overcome in future by use of UHPLC technique.

## Supporting information

**S1 Table. Solution stability favipiravir.**
(DOCX)

**S2 Table. Solution stability remdesivir.**
(DOCX)

**S3 Table. Linearity test.**
(DOCX)

**S4 Table. Precision favipiravir analyst 01.**
(DOCX)

**S5 Table. Precision remdesivir analyst 01.**
(DOCX)

**S6 Table. Precision favipiravir analyst 02.**
(DOCX)

**S7 Table. Precision remdesivir analyst 02.**
(DOCX)

**S8 Table. Accuracy favipiravir.**
(DOCX)

**S9 Table. Accuracy remdesivir.**
(DOCX)

**S10 Table. Robustness favipiravir.**
(DOCX)

**S11 Table. Robustness remdesivir.**
(DOCX)

**S12 Table. Stress degradation favipiravir.**
(DOCX)

**S13 Table. Stress degradation remdesivir.**
(DOCX)

**S14 Table. System suitability favipiravir.**
(DOCX)

**S15 Table. System suitability remdesivir.**
(DOCX)

## Acknowledgments

The authors, also, acknowledge with thanks Novamed Pharmaceuticals (Pvt.) Ltd. Lahore-Pakistan, for the kind gift of Favipiravir (API), Remdesivir GS-621763, and other excipients used in formulation of coloaded capsule for successful completion of this study.

## Author contributions

**Conceptualization:** Barkat Ali Khan.

**Data curation:** Muhammad Iqbal.

**Formal analysis:** Muhammad Iqbal.

**Investigation:** Nasr Ullah Khan.

**Software:** Muhammad Iqbal.

**Supervision:** Barkat Ali Khan.

**Writing – original draft:** Muhammad Iqbal.

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
