## [Decision Letter · Decision Letter 0]

2 Feb 2025

PONE-D-25-03276A SYSTEMIC APPROACH TO ESTIMATE AND VALIDATE RP-HPLC ASSAY METHOD FOR REMDESIVIR AND FAVIPIRAVIR IN CAPSULE DOSAGE FORMPLOS ONE

Dear Dr. Khan,

Thank you for submitting your manuscript to PLOS ONE. After careful consideration, we feel that it has merit but does not fully meet PLOS ONE’s publication criteria as it currently stands. Therefore, we invite you to submit a revised version of the manuscript that addresses the points raised during the review process.

We look forward to receiving your revised manuscript.

Kind regards,

Abdelwahab Omri, Pharm B, Ph.D, Laurentian University

Academic Editor

PLOS ONE

Journal Requirements:

2. In the online submission form, you indicated that “Data id available from corresponding author on reasonable request”

4.  Please ensure that you refer to Figure 1-3 in your text as, if accepted, production will need this reference to link the reader to the figure.

5. We note you have included a table to which you do not refer in the text of your manuscript. Please ensure that you refer to Table 1-7 in your text; if accepted, production will need this reference to link the reader to the Table.

Reviewers' comments:

Reviewer's Responses to Questions

**Comments to the Author**

1. Is the manuscript technically sound, and do the data support the conclusions?

Reviewer #1: Partly

Reviewer #2: Yes

Reviewer #3: Yes

Reviewer #4: No

2. Has the statistical analysis been performed appropriately and rigorously? 

Reviewer #1: No

Reviewer #2: Yes

Reviewer #3: Yes

Reviewer #4: No

3. Have the authors made all data underlying the findings in their manuscript fully available?

Reviewer #1: Yes

Reviewer #2: Yes

Reviewer #3: Yes

Reviewer #4: No

4. Is the manuscript presented in an intelligible fashion and written in standard English?

Reviewer #1: No

Reviewer #2: Yes

Reviewer #3: Yes

Reviewer #4: No

5. Review Comments to the Author

Reviewer #1: In Assay method for dual drug combination by HPLC method is lacking novely and new finding.

Chromatogram of blank, specificity, and LLQ is missing

Without any innovative steps and methods like QbD, greenness, and chemometrics, it seems simple experimental methods. For a research article, the author should incorporate new methodology, processes, and findings that add value to existing research.

Reviewer #2: 1 References are cited as numbers superscripts, however are listed alphabetically , revise as per journal guidelines

2. Table no.7 degree Celsius sign missing

3. Elaborate on significance of degradation studies in discussion

Reviewer #3: Correct manuscript for grammatical mistakes.

Write complete source information for instruments, materials and software used in the proposed research work.

Format references properly as per guidelines.

The proposed method can be evaluated for greenness and whiteness profile.

For greenness and whiteness profile assessment of the proposed method, the authors can cite following recently published articles;

https://doi.org/10.1093/jaoacint/qsad013,
https://doi.org/10.1007/s42250-024-01007-z,
https://doi.org/10.1016/j.scp.2024.101523,
https://doi.org/10.1093/chromsci/bmad054,
https://doi.org/10.1093/jaoacint/qsad108

Reviewer #4: Reviewer Comments for Authors

Abstract: References should not be included in the abstract.

Mobile Phase Composition: It is preferable to state the mobile phase composition in terms of ratios rather than actual volumes.

Run Time Justification: A justification is needed for the extended run time for the separation of only two drugs.

Linearity Range & LOQ: Justify the selected linearity range for both drugs. Additionally, discrepancies between the linearity range and LOQ should be clarified.

Novelty Justification: The manuscript claims novelty, but several HPLC methods have been reported in the literature for this combination. The authors should justify why their method is unique compared to existing ones (e.g., references: https://analyticalsciencejournals.onlinelibrary.wiley.com/doi/10.1002/cem.3548?af=R,
https://pmc.ncbi.nlm.nih.gov/articles/PMC10203959/).

Drug Purity: The purity of the drugs used should be mentioned in the materials section.

Methodology Heading (Section 2.4): The heading "Procedure" is vague. It should clearly define the purpose of the procedure.

Target Concentration Justification: The rationale for selecting target concentrations must be provided.

Incomplete Units: Some values in the methodology section are missing units. Ensure all numerical values have appropriate units.

Precision Studies: Percentage relative standard deviation (%RSD) should be reported for precision studies.

Robustness Studies: The robustness table should be revised for clarity, showing parameter variations and their impact more effectively.

Quality by Design (QbD) Application: The abstract mentions the application of QbD, but there is no detailed discussion on risk assessment, DOE application, MODR, or model validation in the results and discussion section. This needs to be addressed.

6. PLOS authors have the option to publish the peer review history of their article (what does this mean? ). If published, this will include your full peer review and any attached files.

**Do you want your identity to be public for this peer review?** For information about this choice, including consent withdrawal, please see our Privacy Policy .

Reviewer #1: No

Reviewer #2: No

Reviewer #3: No

Reviewer #4: No

---

## [Author Response · Author response to Decision Letter 1]

19 Feb 2025

1In Assay method for dual drug combination by HPLC method is lacking novely and new finding.

Rep: Yes, it is a novel HPLC method in sense of composition of mobile phase and column as well as 1st time coloaded in oral capsule formulation.

2Chromatogram of blank, specificity, and LOQ is missing

Rep:: Chromatographs of Blank, Specificity added in Specificity test, LOQ was determined from Slope of Response as per ICH guidelines.

3Without any innovative steps and methods like QbD, greenness, and chemometrics, it seems simple experimental methods. For a research article, the author should incorporate new methodology, processes, and findings that add value to existing research.

Rep: Green chemistry discussed in introduction part and included in conclusion part as per instructions of reviewer.

1References are cited as numbers superscripts, however are listed alphabetically, revise as per journal guidelines

Rep: Reference cited style corrected to “Vancouver” as per reviewer comments and journal guidelines.

2Table no.7-degree Celsius sign missing

Rep: Corrected as instructed by the reviewer

3Elaborate on significance of degradation studies in discussion

Rep: Significance of degradation studies added in 3.8 Stress degradation as per reviewer instructions.

1Correct manuscript for grammatical mistakes.

Rep: Corrected as instructed by the reviewer

2Write complete source information for instruments, materials and software used in the proposed research work.

Rep: Source information and software version added for customized instruments as per reviewer instructions.

3Format references properly as per guidelines.

Rep: Corrected as instructed by the reviewer

4The proposed method can be evaluated for greenness and whiteness profile.

Rep: Green chemistry discussed in introduction part and included in conclusion part as per instructions of reviewer.

1Abstract: References should not be included in the abstract.

Rep: According to the reviewer instructions, references are removed from abstract.

2Mobile Phase Composition: It is preferable to state the mobile phase composition in terms of ratios rather than actual volumes.

Rep: Corrected as instructed by the reviewer

3Run Time Justification: A justification is needed for the extended run time for the separation of only two drugs.

Rep: The extended run time was due to remdesivir peak behavior, which shows inconsistent behavior in elution pattern.

4Linearity Range & LOQ: Justify the selected linearity range for both drugs. Additionally, discrepancies between the linearity range and LOQ should be clarified.

Rep: Corrected as instructed by reviewer to know the response is linear (proportional to concentration), or the response may be non-linear, especially at higher concentrations.

5 Novelty Justification: The manuscript claims novelty, but several HPLC methods have been reported in the literature for this combination. The authors should justify why their method is unique compared to existing ones (e.g., references: https://analyticalsciencejournals.onlinelibrary.wiley.com/doi/10.1002/cem.3548?af=R,
https://pmc.ncbi.nlm.nih.gov/articles/PMC10203959/).

Rep: Yes, it is a novel HPLC method in sense of composition of mobile phase and column as well as 1st time coloaded in oral capsule formulation and described in introduction section.

5Drug Purity: The purity of the drugs used should be mentioned in the materials section.

Rep: Drug purity has been mentioned as per instructions of reviewer in material section.

6Methodology Heading (Section 2.4): The heading "Procedure" is vague. It should clearly define the purpose of the procedure.

Rep: Procedure has been replaced with Experimental methodology as per instructions of reviewer.

7Target Concentration Justification: The rationale for selecting target concentrations must be provided.

Rep: The rationale for target concentration has been describe in experimental methodology as per instructions of reviewer.

9 Incomplete Units: Some values in the methodology section are missing units. Ensure all numerical values have appropriate units.

Rep: Corrected as instructed by the reviewer

10Precision Studies: Percentage relative standard deviation (%RSD) should be reported for precision studies.

Rep: % RSD has been corrected as per instructions of reviewer.

11Robustness Studies: The robustness table should be revised for clarity, showing parameter variations and their impact more effectively.

Rep: Robustness studies has been elaborated for variable parameters and their impact as per instructions of reviewer.

12Quality by Design (QbD) Application: The abstract mentions the application of QbD, but there is no detailed discussion on risk assessment, DOE application, MODR, or model validation in the results and discussion section. This needs to be addressed.

Rep: QbD has been added in results section with risk assessment, DOE and validation aspects as per reviewer instructions.

1.Lack of Justification for Key Parameters: The manuscript does not justify the long run time, target concentration selection, or discrepancies in LOQ and linearity.

Rep: The extended run time was due to remdesivir peak behavior, which shows inconsistent behavior in elution pattern.

2QbD Application Not Detailed: Although QbD is mentioned in the abstract, the manuscript does not discuss key elements like risk assessment, design of experiments (DOE), or model validation.

Rep: QbD has been added in results section with risk assessment, DOE and validation aspects as per reviewer instructions.

3Novelty Not Justified: The study does not establish how this method differs significantly from previously reported methods.

Rep: Yes, it is a novel HPLC method in sense of composition of mobile phase and column as well as 1st time coloaded in oral capsule formulation and described in introduction section.

4Formatting & Language Issues: There are inconsistencies in terminology, missing units, and vague methodology headings.

Rep:Terminologies changed, missing units added and vague methodology headings corrected as instructed by the reviewer

5Inadequate Robustness & Precision Reporting: The robustness table needs better organization, and precision studies should include %RSD.

Rep: % RSD has been corrected as per instructions of reviewer in precision test. Robustness studies has been elaborated for variable parameters and their impact as per instructions of reviewer.

---

## [Editor Report · Decision Letter 1]

23 Feb 2025

PONE-D-25-03276R1A SYSTEMIC APPROACH TO ESTIMATE AND VALIDATE RP-HPLC ASSAY METHOD FOR REMDESIVIR AND FAVIPIRAVIR IN CAPSULE DOSAGE FORMPLOS ONE

Dear Dr. Khan,

Thank you for submitting your manuscript to PLOS ONE. After careful consideration, we feel that it has merit but does not fully meet PLOS ONE’s publication criteria as it currently stands. Therefore, we invite you to submit a revised version of the manuscript that addresses the points raised during the review process.

We look forward to receiving your revised manuscript.

Kind regards,

Abdelwahab Omri, Pharm B, Ph.D

Academic Editor

PLOS ONE

Journal Requirements:

**Additional Editor Comments:**

Dear Dr. Khan,

Thank you for submitting the revised manuscript titled "A Systemic Approach to Estimate and Validate RP-HPLC Assay Method for Remdesivir and Favipiravir in Capsule Dosage Form" to PLOS ONE. We appreciate the efforts you and your co-authors have made in addressing the reviewers’ comments. After careful evaluation of your revisions, we find that the manuscript has significantly improved. However, a few minor concerns need to be addressed before we can proceed with final acceptance.

Required Revisions:

1. Explicit Comparison with Existing Methods & Justification of Uniqueness:

o While the manuscript claims novelty based on mobile phase composition and co-loading in oral capsules, it does not critically compare the proposed method with existing RP-HPLC methods.

o Please provide a detailed comparison with prior studies, highlighting the advantages of your method in terms of sensitivity, specificity, robustness, or applicability. A direct comparison table or a critical discussion in the introduction or discussion section would be beneficial.

2. Details on Risk Assessment, DOE Application, MODR, and Model Validation:

o The manuscript briefly mentions quality by design (QbD) and risk assessment but does not provide sufficient details on method operable design region (MODR) and model validation.

o Please include a structured discussion explaining the specific risk factors considered, how DOE (Design of Experiments) was applied, and how model validation was performed. Adding supporting data would enhance clarity.

3. Justification for Extended Run Time:

o The explanation provided for the extended run time due to remdesivir’s inconsistent elution pattern is acceptable. However, including chromatographic overlays or additional discussion on why shorter run times were not feasible would further strengthen the justification.

4. Explanation of LOQ and Linearity Discrepancies:

o The manuscript presents LOQ and linearity values but does not explain why any potential discrepancies might exist. If non-linearity was observed at higher concentrations, please clarify how the method ensures accuracy within the tested range.

---

## [Author Response · Author response to Decision Letter 2]

4 Mar 2025

Query: Explicit Comparison with Existing Methods & Justification of Uniqueness:

While the manuscript claims novelty based on mobile phase composition and co-loading in oral capsules, it does not critically compare the proposed method with existing RP-HPLC methods.

Please provide a detailed comparison with prior studies, highlighting the advantages of your method in terms of sensitivity, specificity, robustness, or applicability. A direct comparison table or a critical discussion in the introduction or discussion section would be beneficial.

Response: A detailed comparison of existing papers, and our paper uniqueness has been incorporated in introduction section as per instructions of reviewer. It can be traced as red color.

Query: Details on Risk Assessment, DOE Application, MODR, and Model Validation:

The manuscript briefly mentions quality by design (QbD) and risk assessment but does not provide sufficient details on method operable design region (MODR) and model validation.

Please include a structured discussion explaining the specific risk factors considered, how DOE (Design of Experiments) was applied, and how model validation was performed. Adding supporting data would enhance clarity.

Response: A detailed method operable design region and DOE has been included in introduction section as per instructions of reviewer.

Query: Justification for Extended Run Time:

The explanation provided for the extended run time due to remdesivir’s inconsistent elution pattern is acceptable. However, including chromatographic overlays or additional discussion on why shorter run times were not feasible would further strengthen the justification.

Response: Explanation has been added in LOD and LOQ parameters for non linear response and ensuring results in accuracy, precision, robustness as per instructions of reviewer.

---

## [Editor Report · Decision Letter 2]

7 Mar 2025

A SYSTEMIC APPROACH TO ESTIMATE AND VALIDATE RP-HPLC ASSAY METHOD FOR REMDESIVIR AND FAVIPIRAVIR IN CAPSULE DOSAGE FORM

PONE-D-25-03276R2

Dear Dr. Barkat Ali Khan,

We’re pleased to inform you that your manuscript has been judged scientifically suitable for publication and will be formally accepted for publication once it meets all outstanding technical requirements.

Kind regards,

Abdelwahab Omri, Pharm B, Ph.D, Laurentian University, Canada

Academic Editor

PLOS ONE

---

## [Editor Report · Acceptance letter]

PONE-D-25-03276R2

PLOS ONE

Dear Dr. Khan,

I'm pleased to inform you that your manuscript has been deemed suitable for publication in PLOS ONE. Congratulations! Your manuscript is now being handed over to our production team.

Kind regards,

on behalf of

Dr. Abdelwahab Omri

Academic Editor

PLOS ONE